

**PeerJ Hubs**
In partnership with

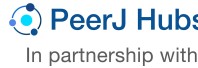
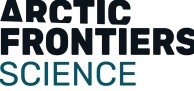

# Moving beyond the physical impervious surface impact and urban habitat fragmentation of Alaska: quantitative human footprint inference from the first large scale 30 m high-resolution Landscape metrics big data quantification in R and the cloud

Moriz Steiner and Falk Huettmann

-EWHALE lab-, Biology and Wildlife Department, Institute of Arctic Biology, University of Alaska Fairbanks (UAF), Fairbanks, AK, United States

Corresponding author
Moriz Steiner, msteiner2@alaska.edu

## ABSTRACT

With increased globalization, man-made climate change, and urbanization, the landscape–embedded within the Anthropocene-becomes increasingly fragmented. With wilderness habitats transitioning and getting lost, globally relevant regions considered 'pristine', such as Alaska, are no exception. Alaska holds 60% of the U.S. National Park system's area and is of national and international importance, considering the U.S. is one of the wealthiest nations on earth. These characteristics tie into densities and quantities of human features, *e.g.*, roads, houses, mines, wind parks, agriculture, trails, *etc.*, that can be summarized as 'impervious surfaces.' Those are physical impacts and actively affecting urban-driven landscape fragmentation. Using the remote sensing data of the National Land Cover Database (NLCD), here we attempt to create the first quantification of this physical human impact on the Alaskan landscape and its fragmentation. We quantified these impacts using the well-established landscape metrics tool 'Fragstats', implemented as the R package "landscapemetrics" in the desktop software and through the interface of a Linux Cloud-computing environment. This workflow allows for the first time to overcome the computational limitations of the conventional Fragstats software within a reasonably quick timeframe. Thereby, we are able to analyze a land area as large as approx. 1,517,733 $km^2$ (state of Alaska) while maintaining a high assessment resolution of 30 m. Based on this traditional methodology, we found that Alaska has a reported physical human impact of c. 0.067%. We additionally overlaid other features that were not included in the input data to highlight the overall true human impact (*e.g.*, roads, trails, airports, governance boundaries in game management and park units, mines, *etc.*). We found that using remote sensing (human impact layers), Alaska's human impact is considerably underestimated to a meaningless estimate. The state is more seriously fragmented and affected by humans than commonly assumed. Very few areas are truly untouched and display a high patch density with corresponding low mean patch sizes throughout the study area. Instead, the true human impact is likely close to 100% throughout Alaska for several metrics. With

these newly created insights, we provide the first state-wide landscape data and inference that are likely of considerable importance for land management entities in the state of Alaska, and for the U.S. National Park systems overall, especially in the changing climate. Likewise, the methodological framework presented here shows an Open Access workflow and can be used as a reference to be reproduced virtually anywhere else on the planet to assess more realistic large-scale landscape metrics. It can also be used to assess human impacts on the landscape for more sustainable landscape stewardship and mitigation in policy.

## INTRODUCTION

Considering global and universal governance, we live in the Anthropocene, now often referred to as the Pyrocene. At the same time, man-made $CO_2$ release and physical impact remain widely unabated and increase ('the Acceleration', see *Colvile, 2016*) to a degree never witnessed before. This is a result of human-accelerated climate change and its impacts on the northern ecosystem (*Koenigk, Key & Vihma, 2020*; *Prowse et al., 2006*; *Reist et al., 2006*). An acknowledgment and gaining an increased understanding of the landscape humans live in has been of interest to humankind for a long time but remains widely unachieved. In the early days of humankind, that might have been to locate suitable areas for the construction of shelters or to better understand where certain crops and plants grow or where prey is more likely to be found (*Barker, 2006*; *Hoffecker, 2005*). In modern days, such questions are likely less important to an individual but to wider governance entities and national setups (*e.g.*, *Koohafkan, 2000*); nonetheless, gaining a greater understanding of our landscape–lebensraum–and its sustainability is still a very relevant objective (see *Leopold, 2017*; *Nelson, 1998*). Nowadays, on a global scale, one mainly tries to use the latest computer technology and satellites to obtain a comprehensive synergy and data about the landscape we live in for effective sustainability progress (*Frohn, 2018*; *Groom et al., 2006*; *Jorgenson & Grosse, 2016*). Such pixel-based information is of crucial importance to land, landscape, and biodiversity managers, conservation entities, as well as research institutions-using public resources-now and for the future (*Pettorelli, Safi & Turner, 2014*).

To analyze the structure of the landscape and its characteristics, landscape metrics analyses are of extremely high importance (see *Nowosad & Stepinski, 2019*) while public tools exist already for decades, a rather widely-used one is Fragstats (*McGarigal, 1995*). Such landscape analyses have also already been performed for Alaska, but only in part, *e.g.*, see an analysis for the Glacier Bay National Park in *Klaar, Maddock & Milner (2009)*, for the North Slope of Alaska in *Naito (2014)*, or Tanana Valley State Forest in *Steiner & Huettmann (2023)*. However, to our knowledge, previous studies either focused on a smaller area within Alaska and remained with narrow metrics (see *Hesselbarth et al., 2019* for over 100 metrics options) and, therefore, never assessed the state as a whole in synergy

or just utilized a coarse pixel size. Outside of Alaska, landscape metrics have been widely used to assess landscapes. However, they often just used minuscule study areas (mostly under 1,000 km$^2$) compared to the whole state of Alaska (1,517,733 km$^2$), due to the data volume and scale limitations (see *Kubacka & Piniarski, 2024*; *Lausch & Herzog, 2002*; *Masoudi, Richards & Tan, 2024*; *Petrosyan & Karathanassi, 2011* and references therein). Despite Alaska's Indigenous history and perspective, holistic and wider high-resolution quantitative summary products are still missing, not allowing for a modern state-wide governance of the human footprint overall.

Thus far, the standard analysis for landscape metrics is still 'aspatial', meaning that they provide insights into the landscape structure in the form of a summary table but without being explicit in space and time and thus carry no map, *e.g.*, in an Open-GIS framework (see QuAntarctica and QGreenland for other Polar regions; *Matsuoka et al., 2021*; *Moon et al., 2022*). Such an approach could start to provide some meaningful insights but has few applications beyond providing an overall summary. It lacks pixel information. To produce a spatial product that can be re-utilized in GIS and further analyzed, a moving window analysis is required instead (*e.g.*, *Hagen-Zanker, 2016*) but is currently hindered by computing power, funding aims, and needed sophistication.

Grasping the overall impact that humans have on the landscape is a hard-to-achieve task and often difficult to truly quantify in a holistic fashion. This is especially true for rather remote wilderness areas like the Arctic or the state of Alaska. In such scenarios, dated remote sensing applications are the prime methodological choice for convenience and coverage (*Zhou, Li & Chen, 2011*). That way, the impact humans have on the landscape can be observed and measured by incorporating a series of artificial features. Generally, features that are considered part of 'impervious surfaces,' *e.g.*, areas built up with asphalt - are rather clear and drastic indicators of human impact (*Li et al., 2011*; *Slonecker, Jennings & Garofalo, 2001*; *Weng, 2012*). Such features are varied and include, for instance, built-up places where forests and watersheds are partially or fully destroyed, and replaced by roads, roofs, houses, trails, mines, wind parks, agricultural fields, sports fields, compacted soils, parking lots, train tracks, pipelines, *etc.*, (*Granato & Northborough, 2015*; *Schueler, 2000*); all else from wild nature, vegetation, and natural resources. Other more indirect human features–widely overlooked but often equally relevant-like culture, National Park and protected land boundaries, governance boundaries, and game management unit boundaries also impact the landscape and its habitat composition and characteristics, *e.g.*, through far-reaching intense management policies. Such impacts often reach beyond any assigned boundaries but changing structure and species compositions (typical example provided with predator control policies–coming with a cultural and national mindset-that affect the food chain at large as well as vegetation) (*Cheţan & Dornik, 2021*; *Kubacka et al., 2022*; *Van Ballenberghe, 2006*).

The U.S. is one of the wealthiest governance entities in the world, with a budget that has global impacts and serves as a governance role model. Already, the money spent on the military is almost twice as much as what every other nation spends or has available to operate based on their GDP (*e.g.*, Wiki on world GDPs: *Wikipedia, 2025*). If wealthy nations have such landscape conservation problems, *e.g.*, loss of wilderness and

fragmentation, most of the world cannot do any better. It is a significant problem to allocate so much money to a military budget. Alaska has many major military bases, airports, and ports, which are strategically relevant for the West. The landscapes are directly affected by it and always have been, considering Alaska is a colonial construct. The relevance of GDP for the landscape is widely recognized as a scientific fact, as expressed in the discipline of Ecological Economics and associated citations (*e.g.*, see *Huettmann, 2014* for Alaska).

Here, we investigate and attempt for the first time to quantify the human impact (imperviousness) on the Alaskan landscape by utilizing remotely sensed data, a diverse and inclusive set of Open Access datasets, modern landscape metrics assessments, and cloud computing, making use of best-available tools and methods. To our knowledge, this is the first state-wide human footprint assessment using landscape metrics in the U.S. and its states (compare with *Kaminski et al., 2021*; *Linke et al., 2005*). Gaining a better understanding of the Alaskan landscape and the impact humans have on it can provide a series of stakeholders with a new tool to better study, manage, and conserve the Alaskan landscape on a pixel level (*Fortin, 1999*; *Gülçin, 2020*). The main aim of this manuscript is to provide a proof-of-concept that such high landscape-scale data volumes can be processed using the currently publicly available 'landscapemetric' algorithm. The framework and workflow provided here can be used as a reference to allow for future applications in other areas of the globe that can benefit from large-scale landscape metric assessments.

## METHODS

We conducted a statewide landscape analysis using ten landscape metrics based on the 30 m resolution National Land Cover Database (https://www.mrlc.gov/data/nlcd-2016-land-cover-alaska) for urban landcover classes. The analysis was carried out in R, using two options, the RStudio desktop software (*Hesselbarth et al., 2019*) and a Linux-powered cloud computing network to quantify the human impact on Alaska's landscape. The main reason for carrying out the analysis on the cloud computer as well as the desktop version of R was to provide an option that is 'free' for a global audience and can be performed by anyone with a PC and internet (desktop option). The other option is more powerful and time-saving but usually not free of charge (cloud computing option, widely restricted for a global audience). To improve the minimum estimate of the human impact on Alaska's landscape, we additionally included other Open Access GIS layers in our final results, featuring direct and indirect effects of the human footprint on the landscape. This allows for a more holistic and meaningful analysis toward the understanding of the true human impact for policy changes.

### Study area

Alaska is located on the far northwestern end of North America, with the northernmost point of the United States (71°23′20″N), the westernmost point of the continental United States (173°11′10″E). The western southern limit of Alaska is 51°15′44″N, and the eastern southern limit is 54°39′44″N. An overview of the study area (Alaska) in the global context

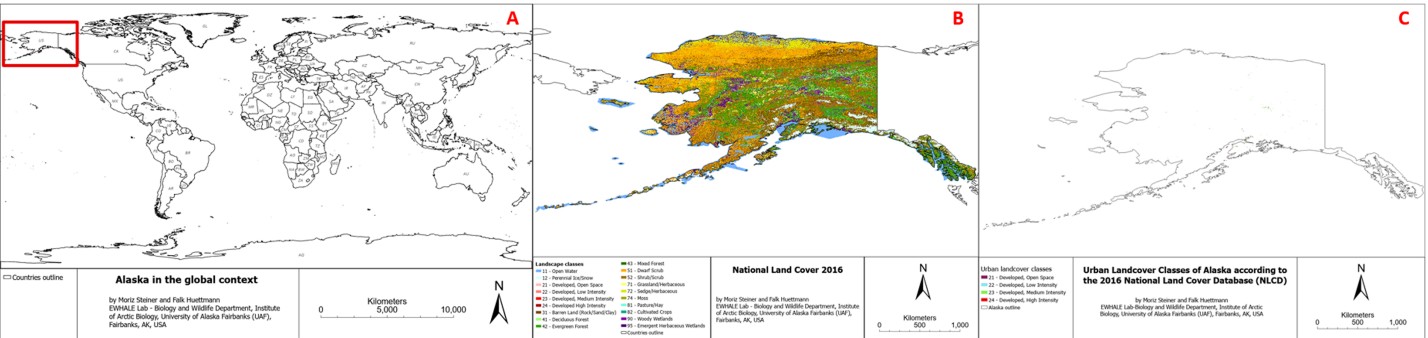

**Figure 1** (A) Study area in the global context, (B) Study area overview with underlying landcover classes (source: https://www.mrlc.gov/data/nlcd-2016-land-cover-alaska), (C) Urban landcover classes of Alaska according to the 2016 national land cover data.

can be found in Fig. 1A. The climate in Alaska highly varies, mostly depending on the proximity to the sea and the latitude, allowing for climate class ranges from arctic to maritime (*Shulski & Wendler, 2007*). Average temperatures in the state of Alaska range from 5.5 °C in the South to about −12 °C in the North, and extreme temperatures were recorded as low as −62 °C and as high as 38 °C (*WRCC, 2024*). Alaska is home to some of the most extreme mountain ranges in North America, with the St. Elias Mountains, the Chugach Range, and the tallest mountain in North America (Denali–height of 6,190 m) (*Wahrhaftig, 1965*). Alaska is also known for its vast tundra and taiga landscapes with an uncountable number of lakes and is characterized by almost 40% of near-surface permafrost (*Bonan, Chapin & Thompson, 1995*; *Pastick et al., 2015*).

Alaska has a population of approximately 735,000 people (as of 2024), with a median age of 35.3 years, a median income of 86,000 USD, and a poverty rate of 10.5% (additional socio-economic details about the study area can be found here: https://datausa.io/profile/geo/alaska).

Alaska has a long human (Indigenous) history and is located within that North American setting; it features rich natural resources (*Huryn & Hobbie, 2012*; *Naske & Slotnick, 2014*; *Truett & Johnson, 2000*) and the vast majority of the U.S. National Park System and protected land–a highly prized land entity–within (*Bigelow & Borchers, 2017*).

Figure 1 provides an overview of the study area (the state of Alaska) and the underlying land use classes in the National Land Cover Database (NLCD) map, of which we used all "Developed, …" ones (21, 22, 23, 24). It additionally includes the landcover classes description in the legend, and more information on the utilized data can be found in the 'Data' section.

## Data

For this project, we used the 2016 30 × 30 m resolution landcover map of Alaska, provided by the National Land Cover Database (NLCD) (see https://www.mrlc.gov/data/nlcd-2016-land-cover-alaska). This map product represents the national standard of land cover maps for the largest state of the US, given its source (U.S. Geological Survey (USGS) Earth Resources Observation and Science (EROS) Center). The downloaded raw data comes as a

.img file, which we converted in ArcGIS Pro (optionally can also be done in OpenGIS (QGIS)) into a usable .tiff file, using the 'Resample' tool, which carries a total file size of 8,269.77 MB (approx. 8.2 GB). It is an eight-bit unsigned file with a total of 124,236 columns and 67,844 rows, using the WGS 84 geographic projection (WGS_1984_Albers- using 'meters' as the linear unit), covering the entirety of Alaska, US.

We chose this dataset because it is the official national map product of Alaska's landcover classes. It can be easily downloaded onto a personal computer as a single file from the internet free of charge (open-source) and has a high resolution of 30 × 30 m.

This pixel dataset easily misses linear features and comes with a total of 20 categories, which hardly represent the wide spectrum of landscapes present in Alaska (see legend and class descriptions online: https://www.mrlc.gov/data/legends/national-land-cover-database-class-legend-and-description). To specifically analyze the direct physical impact of humans on the landscape of Alaska, we used the urban classes: "21", "22", "23", and "24", corresponding to "Developed, Open Space", "Developed, Low Intensity", "Developed, Medium Intensity", and "Developed, High Intensity" respectively. These categories were merged and considered as the urban areas of the dataset, which can be seen in Fig. 1B and highlighted in Fig. 1C.

### Cumulative human impact layers

To provide insights into the actual human impact on the Alaskan landscape, we compiled a set of human impact layers. This set of layers aims to provide an overview of the (actual) human impact rather than the seemingly underestimated urban areas outlined by the National Land Cover Database (NLCD). For each of these layers, we utilized the most recent/updated version that is publicly available. These layers are described in Table 1, with their corresponding meanings and sources.

### Tools

Given that the most-used software for the indicated objective does not directly allow for such a proposed landscape analysis on a state-wide level due to input data size limits of approximately 50 MB (*McGarigal, 2015*), we attempted to compile the results using a few different options. Initially, we attempted several in-built tools in ArcGIS Pro, ArcGIS, and QGIS (including GRASS GIS). However, none of these delivered meaningful results due to a variety of reasons, *e.g.*, discontinued tools, configuration issues, and computational size issues. GRASS GIS, with its built-in landscape analysis tools, also presents limitations in terms of the configurational errors it produces, which constrain the use of those tools. ArcGIS and ArcGIS Pro discontinued previously existing tools and plug-ins to analyze landscape metrics, limiting their use to the most recent versions. The only software allowing for the successful creation of landscape metrics was R and RStudio (version 3.3), allowing us to replicate the results Fragstats would provide. We utilized the R package 'landscapemetrics' (https://r-spatialecology.github.io/landscapemetrics/-*Hesselbarth et al., 2019*). The code we used to analyze the landscape metrics can be found in Appendix 1 (https://scholarworks.alaska.edu/handle/11122/15662). This code was used for both the desktop-based RStudio software as well as the R session in the Linux cloud-computing

**Table 1 Cumulative human impact layers.**

| Layer name | Physical (direct) | Management (indirect) | Meaning | Impact on the landscape | Data provider and year of latest update | Source |
|---|---|---|---|---|---|---|
| Roads | Yes | | Major roadways, mostly paved. | | Alaska Department of Transportation and Public Facilities (2024) | https://data-soa-akdot.opendata.arcgis.com/search?tags=roads |
| Road segments | Yes | | Major roadways | Contributes to habitat fragmentation | U.S. Geological Survey (2024) | https://thor-f5.er.usgs.gov/ngtoc/metadata/waf/transportation/ntd/shapefile/TRAN_Alaska_State_Shape.xml (accessed on 15 January 2025) |
| Trail segments | Yes | | Major trails | | U.S. Geological Survey (2024) | https://thor-f5.er.usgs.gov/ngtoc/metadata/waf/transportation/ntd/shapefile/TRAN_Alaska_State_Shape.xml (accessed on 15 January 2025) |
| Bridges | Yes | | Bridges along roadways | Allows for human travel across rivers and can source introduced species into land and waterways | Alaska Department of Transportation and Public Facilities (2024) | https://data-soa-akdot.opendata.arcgis.com/search?tags=roads |
| Airports | Yes | | Airports | Allows for human access in remote areas (hunting, fishing, camping, littering, oil spills, contaminated sites, etc.) | Alaska Department of Transportation and Public Facilities (2024) | https://data-soa-akdot.opendata.arcgis.com/search?tags=air |
| Airport runways | Yes | | Airport Runways | Contributes to habitat fragmentation and allows for human access in remote areas | U.S. Geological Survey (2024) | https://thor-f5.er.usgs.gov/ngtoc/metadata/waf/transportation/ntd/shapefile/TRAN_Alaska_State_Shape.xml (accessed on 15 January 2025) |
| Mining sites (mineral order and mining prospects) | Yes | | Active and prospecting mining sites | Contributions to resource extraction and habitat degradation are often centers of contaminated sites. Need human access, mostly by roads. | State of Alaska Department of Natural Resources (2024) | https://mapper.dnr.alaska.gov/ |
| Wind turbines (wind parks) | Yes | | Singular wind turbines and wind parks (in accumulation of turbines) | Need human access and disturb flyways (mainly for birds). | State of Alaska Department of Commerce, Community, & Economic Development (2024) | https://gis.data.alaska.gov/datasets/1d70f4ee61f749c28bc80735e4b108b_0/explore |
| National park boundaries | | Yes | United States (Alaska) National Park boundaries | It affects wildlife regulations, hunting harvest regulations, and tourist access and requires park maintenance access. | Alaska Department of Environmental Conservation AGO (2020) | https://gis.data.alaska.gov/datasets/ADEC::alaska-national-parks-preserves-monuments/explore?location=1.069219%2C0.000000%2C1.1 |

(Continued)

| Layer name | Physical (direct) | Management (indirect) | Meaning | Impact on the landscape | Data provider and year of latest update | Source |
|---|---|---|---|---|---|---|
| Game management subunit | | Yes | Alaska Game Management Subunit | Affects wildlife regulations and hunting harvest regulations. | Alaska Department of Fish & Game (2017) | https://hub.arcgis.com/datasets/bcf76d201a2742b187f7014abe49affe_0/explore |
| Livestock districts | Yes | | Established Livestock Districts, not including singular farms | It affects fencing for livestock and wildlife roaming ability and often requires human road access. | State of Alaska Department of Natural Resources (2023) | https://data-soa-dnr.opendata.arcgis.com/datasets/58aef4089a2b402ca3d797704c8a97f8/explore?layer=0&location=0.565104%2C25.999982%2C1.03 |
| Train tracks/Railways | Yes | | Major railways | Contributes to habitat fragmentation | U.S. Geological Survey (2024) | https://thor-f5.er.usgs.gov/ngtoc/metadata/waf/transportation/ntd/shapefile/TRAN_Alaska_State_Shape.xml (accessed on 15 January 2025) |
| Pipelines | Yes | | Major commercial pipelines | Contributes to habitat fragmentation | State of Alaska Department of Natural Resources (2023) | https://gis.data.alaska.gov/datasets/SOA-DNR::pipeline-163360/explore |
| Contaminated sites | Yes | | Active and inactive contaminated sites, including different contaminants | Contaminating the landscape and species therein can lead to significant health declines in affected populations. | Alaska Department of Environmental Conservation AGO (2020) | https://data-soa-adec.opendata.arcgis.com/datasets/ADEC:alaska-dec-contaminated-sites/explore |
| Carbon dioxide ($CO_2$) | | Yes | Sum of atmospheric $CO_2$ over Alaska between June 14th 2024 and June 16th 2024 (mid of last month of dataset compilation). | Contributes to global climate change, altering habitat | Copernicus (2024) | https://browser.dataspace.copernicus.eu/ |
| Methane ($CH_4$) | | Yes | Sum of atmospheric $CH_4$ over Alaska between June 14th 2024 and June 16th 2024 (mid of last month of dataset compilation). | composition and modifying landscape conditions. | Copernicus (2024) | https://browser.dataspace.copernicus.eu/ |
| Alaska native language boundaries | | Yes | Boundaries of Alaska Native languages | Delineates the boundaries of Alaska Native tribes with differences in wildlife management and harvesting practices. | State of Alaska Department of Commerce, Community, & Economic Development (2024) | https://gis.data.alaska.gov/datasets/31758a4be86494f509372aa4373b6d56c_0/explore |

| Layer name | Physical (direct) | Management (indirect) | Meaning | Impact on the landscape | Data provider and year of latest update | Source |
|---|---|---|---|---|---|---|
| Ports and harbors | Yes | | Ports and Harbors | Allows for human access to coastal regions that would otherwise remain wild without road access. | State of Alaska Department of Commerce, Community, & Economic Development (2024) | https://gis.data.alaska.gov/datasets/4ae56eeda3084cbc9ce505912c1beceb_7/explore?location=52.694261%2C29.758243%2C4.97 |
| Mountain management boundaries | | Yes | Major mountain boundaries affecting land management | Affect land management practices. | Global Mountain Biodiversity Assessment (2022) | https://www.earthenv.org/mountains |
| Activities DNR | | Yes | Major activity sites of the Alaska Department of Natural Resources | Indicate major human activity in the landscape, e.g., research stations. | State of Alaska Department of Natural Resources (2024) | https://mapper.dnr.alaska.gov/ |
| Mineral agreement DNR | | Yes | Mineral Agreements reported by the Alaska Department of Natural Resources | Indicate major human mining activity. | State of Alaska Department of Natural Resources (2024) | https://mapper.dnr.alaska.gov/ |
| BLM sites | | Yes | Major activity sites of the Alaska Bureau of Land Management | Indicate major human activity in the landscape, e.g., research stations. | State of Alaska Department of Natural Resources (2024) | https://mapper.dnr.alaska.gov/ |
| BLM other sites | | Yes | Additional activity sites of the Alaska Bureau of Land Management | | State of Alaska Department of Natural Resources (2024) | https://mapper.dnr.alaska.gov/ |

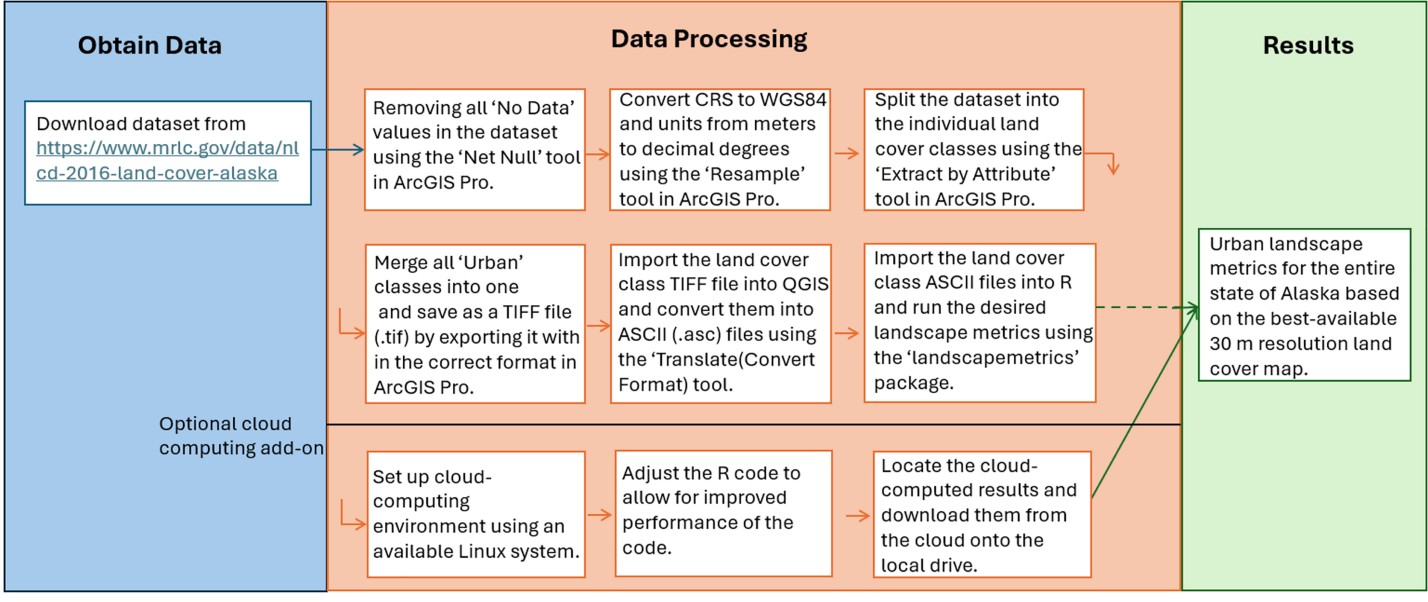

**Figure 2** Workflow for creating spatial landscape metrics in R. 

environment. (The only difference between the code used for the desktop and cloud-computing version was the file directory used to indicate where the data was stored and where the results should be exported).

## Workflow

Our workflow (Fig. 2) starts by downloading public Open Access data (see link above in the 'Data' section), which was then imported into ArcGIS Pro (version 3.2). In ArcGIS Pro, we conducted several steps for data preparation to allow for the proper use of the data. This included the removal of all unused values using the 'Set Null' tool (part of the "Image Analyst" tools), followed by the conversion of the CRS into WGS 84 and setting the units from meters into decimal degrees (EPSG: 4326), to adhere to global CRS standards and streamline the utilized data. To utilize and merge the urban classes, the individual urban classes of the NLDC map were first extracted from the TIFF file using the "Extract by Attribute" tool and were then merged using the "Mosaic To New Raster" tool, both in ArcGIS Pro. This merged urban layer is then imported into open-source QGIS, where it needs to be converted into an ASCII (.asc) file. This resulting ASCII file can then be used for the subsequent moving window analysis in R.

In this study, we analyzed the landscape metrics Coefficient of Variation–Patch Area, Coefficient of Variation–Core Area Index, Mean Core Area, Edge Density, Mean Fractal Dimensional Index, Largest Patch Index, Number Of Patches, Patch Richness, Mean Shape Index, and Total Edge (details of the metrics can be found in the Fragstats manual and online (*McGarigal, 2015*) in a moving window setting. The rationale for focusing on this set of landscape metrics is to include various metrics. An overview of all landscape metrics can be found in Table 2, including the description and function of each metric.

**Table 2 Overview of the included landscape metrics (summarized from *McGarigal, 2015*).**

| Landscape metric name | Description and function |
|---|---|
| Coefficient of variation–Patch area | Measures relative variability of the mean patch size in a landscape. |
| Coefficient of variation–Core area Index | Summarizes the landscape as the coefficient of variation of the core area index (an edge-to-interior ratio) of all patches in the landscape. |
| Mean core area | Refers to the mean of the core areas (the interior area of patches after a user-specified edge buffer is eliminated) across all patches within the assessed landscape. |
| Edge density | Equals the sum of the lengths (m) of all edge segments in the landscape, divided by the total landscape area ($m^2$), multiplied by 10,000 (to convert to hectares). |
| Mean fractal dimensional index | Refers to the mean of the Fractal Dimensional Index (Equals 2 times the logarithm of patch perimeter (m) divided by the logarithm of patch area ($m^2$); the perimeter is adjusted to correct for the raster bias in the perimeter. |
| Largest patch index | Quantifies the percentage of total landscape area comprised by the largest patch. |
| Number of patches | Quantifies the total number of patches in the landscape. |
| Patch richness | Equals the number of patch types (classes) present in the landscape, excluding the landscape border if present. |
| Mean shape index | Equals the sum of the patch perimeter (m) divided by the square root of patch area ($m^2$) for each patch of the corresponding patch type, adjusted by a constant to adjust for a circular standard (vector) or square standard (raster), divided by the number of patches of the same type. |
| Total edge | Equals the sum of the lengths (m) of all edge segments in the landscape. |

The final results were then imported into ArcGIS Pro, where we assigned a coordinate reference system (CRS-WGS 84 EPSG: 4,326) and chose a suitable symbology for visualization. Due to the known high correlation among the computed landscape metrics (*McGarigal, 2015*), and for ease of comprehension, here we only display one of the computed metrics (Coefficient of Variation–Patch Area; that is, the variance of different patch areas in a given pixel as a robust measure of disturbance) in the final overview and its sub-figures in the results section below. All computed landscape metrics can be found as TIFF files in Appendix 2 (https://scholarworks.alaska.edu/handle/11122/15662).

The workflow presented here also contains an optional cloud-computing add-on. This path can be chosen if a Linux cloud computing environment is available, and the results are aimed to be achieved within a shorter timeframe. The only steps that are different for this option are in the final computational section, where the merged ASCII file must be uploaded onto the cloud, where the cloud computer can access it, in addition to a slight code modification accordingly. To make the best use of the improved performance of the cloud computing environment, the code (see Appendix 1-https://scholarworks.alaska.edu/handle/11122/15662) is suggested to be slightly adjusted so the ten different jobs (one for each landscape metric) can be distributed to different cloud computing nodes. Apart from the modification of where the data is stored for the analyses and where the results will be saved, in addition to the optional code optimization, no additional data processing steps are necessary for the cloud-computing option.

For a better overview of the whole study area, we summarized the cumulative human impact on the Alaskan landscape using 1 × 1 decimal degrees hexagons using the "Generate Grids and Hexagons" tool in ArcGIS Pro. For this summary, we used the 'Join
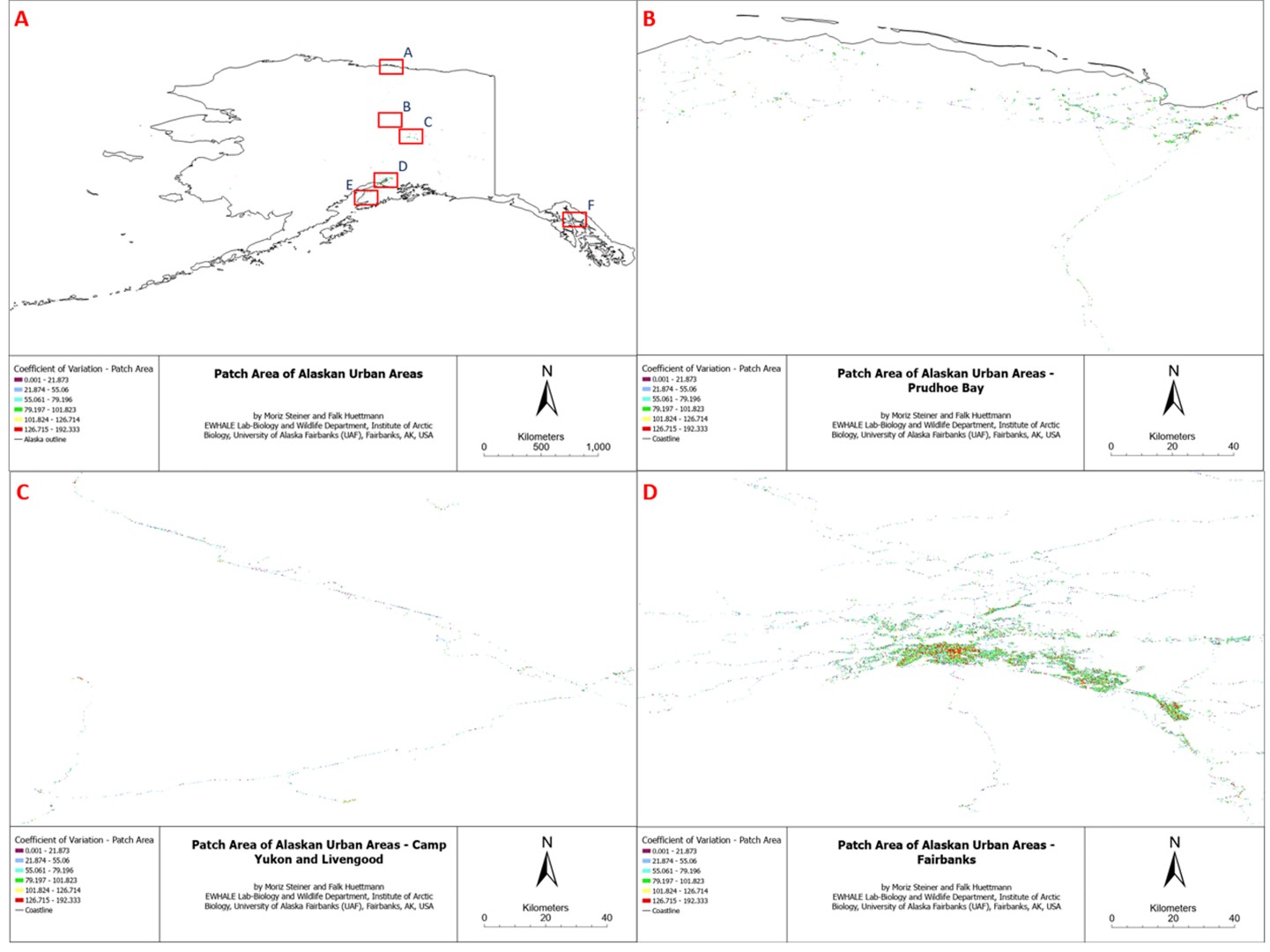

**Figure 3** Patch area coefficient of variation of Alaskan Urban Areas, (B) Prudhoe Bay, (C) Camp Yukon, (D) Fairbanks.

attributes by location' tool in GIS to combine multiple impact layers and their underlying coverage. To further summarize the cumulative impact layers, we created a simple frequency distribution graph in MS Excel with the same color palette as the map overview with hexagons.

## Longitudinal, latitudinal, and altitudinal assessment of urban fragmentation hotspots in Alaska

To identify the urban fragmentation hotspots in Alaska, we overlaid a 2 km lattice grid over the landscape metric and extracted the pixel values. As a result, we obtained 1-degree longitudinal, latitudinal, and altitudinal (meters above sea level) urban fragmentation indices that can be found depicted further below. We used integer degrees on a 'group-by' command to present averages for each degree along the latitude and longitude gradient of

Alaska. With these figures, we attempt to show a profile in the X and Y direction and by altitude to highlight where urban fragmentations can predominately be found in Alaska thus far. This analysis was carried out in GIS and MS Excel and visualized using frequency histograms.

## RESULTS

We attempted to compute ten Alaska-wide landscape metrics for the land cover classes "Developed, Open Space", "Developed, Low Intensity", "Developed, Medium Intensity", and "Developed, High Intensity". Specifically, we attempted to compute the Coefficient of Variation–Patch Area, Coefficient of Variation–Core Area Index, Mean Core Area, Edge Density, Mean Fractal Dimensional Index, Largest Patch Index, Number Of Patches, Patch Richness, Mean Shape Index, and Total Edge for the four merged urban landcover classes. Due to some unknown computational problems in the R package, only two (Coefficient of Variation–Patch Area and Edge Density) of the ten assessed metrics were successfully generated for our study area. The remaining metrics presented the input data as the algorithm result, indicating that some underlying issues did not allow for the successful creation of the landscape metric. Given the visual similarity of the computed landscape metrics, for comprehensive simplicity, here we only display one of the computed metrics (Coefficient of variation of patch area) in the final overview (see Figs. 3A and 4A) and its sub-figures (Figs. 3B–3D and 4B–4D). However, all computed landscape metrics can be found as TIFF files in Appendix 2 (https://scholarworks.alaska.edu/handle/11122/15662) and individual high-resolution figures of all sub-figures of this study in Appendix 4 (https://scholarworks.alaska.edu/handle/11122/15662). The reason for the visual similarity of the results originates from the similarity of the input data (which is the same for all metrics), and all metrics follow similar patterns; for instance, edge density has a proportional relationship with patch area and total edge (*Linke et al., 2005*; *McGarigal, 1995*).

### Direct and indirect human impact components affecting the Alaskan landscape

For the most inclusive overview of the impact that humans have on the modern Alaskan landscape, we additionally overlaid a series of GIS layers indicating direct and indirect impacts on the landscape. Figure 5 aims to visualize the sheer omnipresence and cumulative–true human impact on the landscape of Alaska. As elaborated later, Fig. 5 can still be considered a minimum estimate, as many other direct or indirect factors are either not available online as a digital dataset (*e.g.*, traplines, ATV, powerlines, and snowmobile trails, *etc.*–see Table 3 in the Discussion section) or are highly correlated with other factors (*e.g.*, houses and roofs). The hexagon summary of the cumulative human impacts is presented in Fig. 6, and the simple frequency distribution in Fig. 7.

Figure 5 displays the sum of human impacts on the Alaskan landscape. Figure 6 illustrates the number of impact layers within each hexagon, with red colors indicating a high number of impact layers and dark blue colors indicating a lower number of impact layers. It additionally contains Alaska's National Parks outlined in light blue. Figure 7

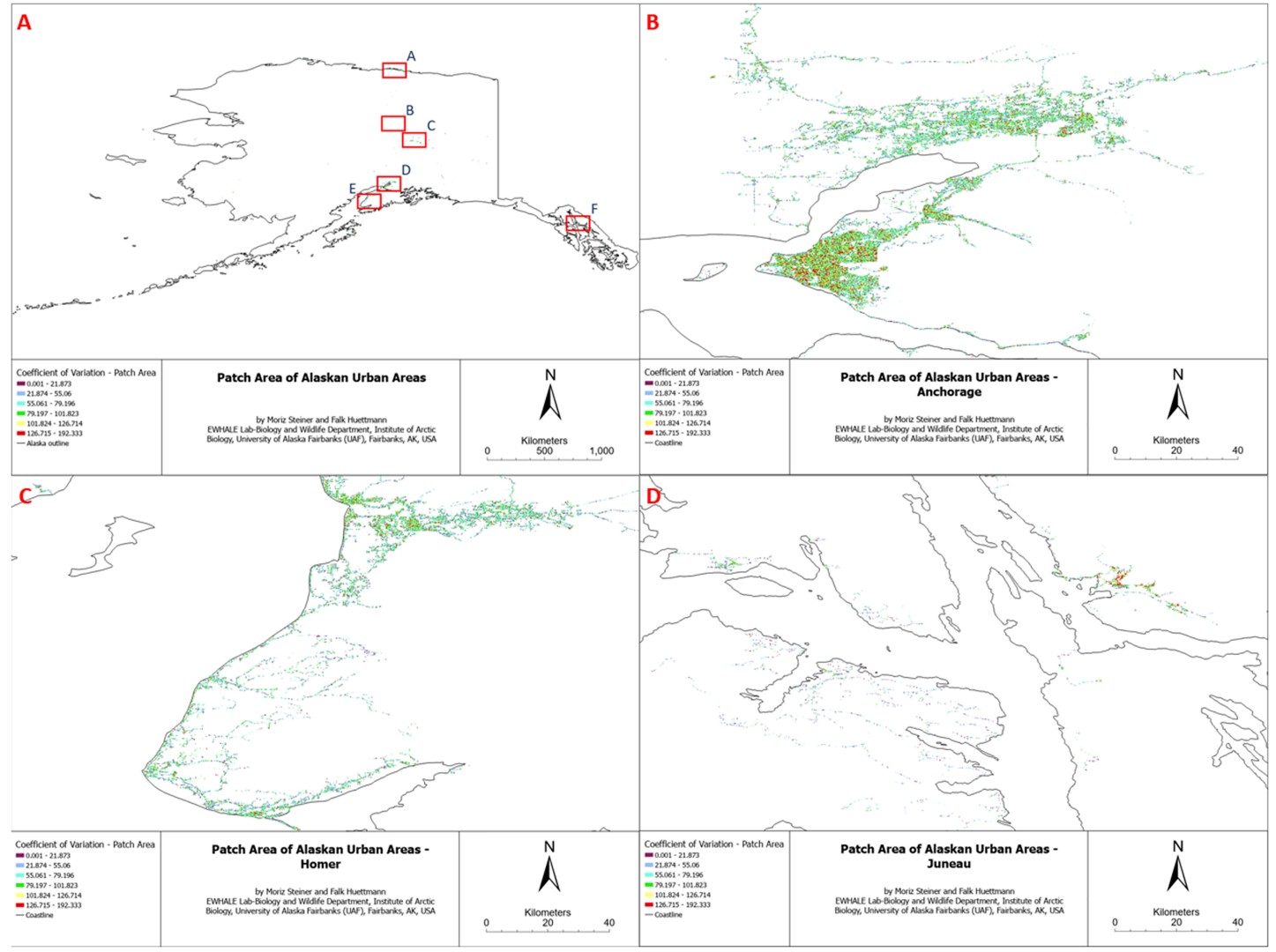

**Figure 4** Patch area coefficient of variation of Alaskan Urban Areas, (B) Anchorage, (C) Homer, (D) Juneau.

illustrates the frequency distribution of the number of human impact layers within one decimal degree hexagons in the state of Alaska, and the color coding is the same as that for Fig. 6. All hexagons with a color other than blue represent areas with more than 11 cumulative impact layers within a 1 × 1 decimal degree area. This means that within each of the non-blue hexagons, more than 11 different layers impact the Alaskan landscape.

## Longitudinal, latitudinal, and altitudinal assessment of urban fragmentation hotspots in Alaska

To outline fragmentation hotspots in Alaska, we also summarized the computed urban area metrics by longitude, latitude, and altitude. The results of that hotspot analysis can be found in Fig. 8 below. These figures aim to display where urban hotspots can be found in Alaska in the landscape and geographic location. It shows the known hotspots like Juneau,

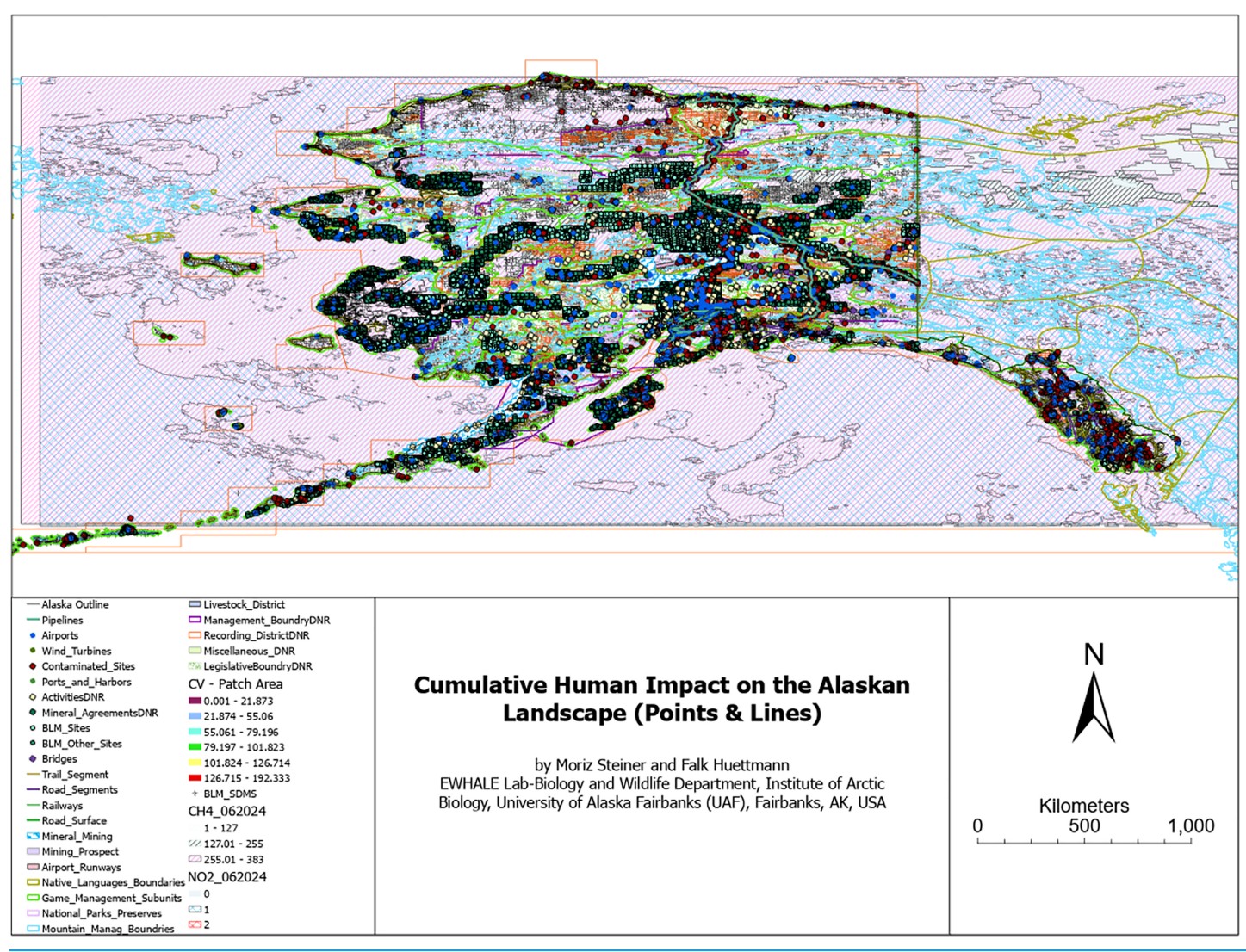

**Figure 5 Cumulative human impact on the Alaskan landscape-points & lines.**

**Table 3 Non-available human impact layers.**

| Layer name | Physical | Management |
|---|---|---|
| Seismic grid lines | Yes | |
| Trails (ATV & Snowmobile) | Yes | |
| Traplines | Yes | |
| Great brown cloud | | Yes |
| Plume from the Norilsk smelter (Asia) | | Yes |
| Climate sound | | Yes |
| Invasive species | | Yes |
| Stocked fish | | Yes |

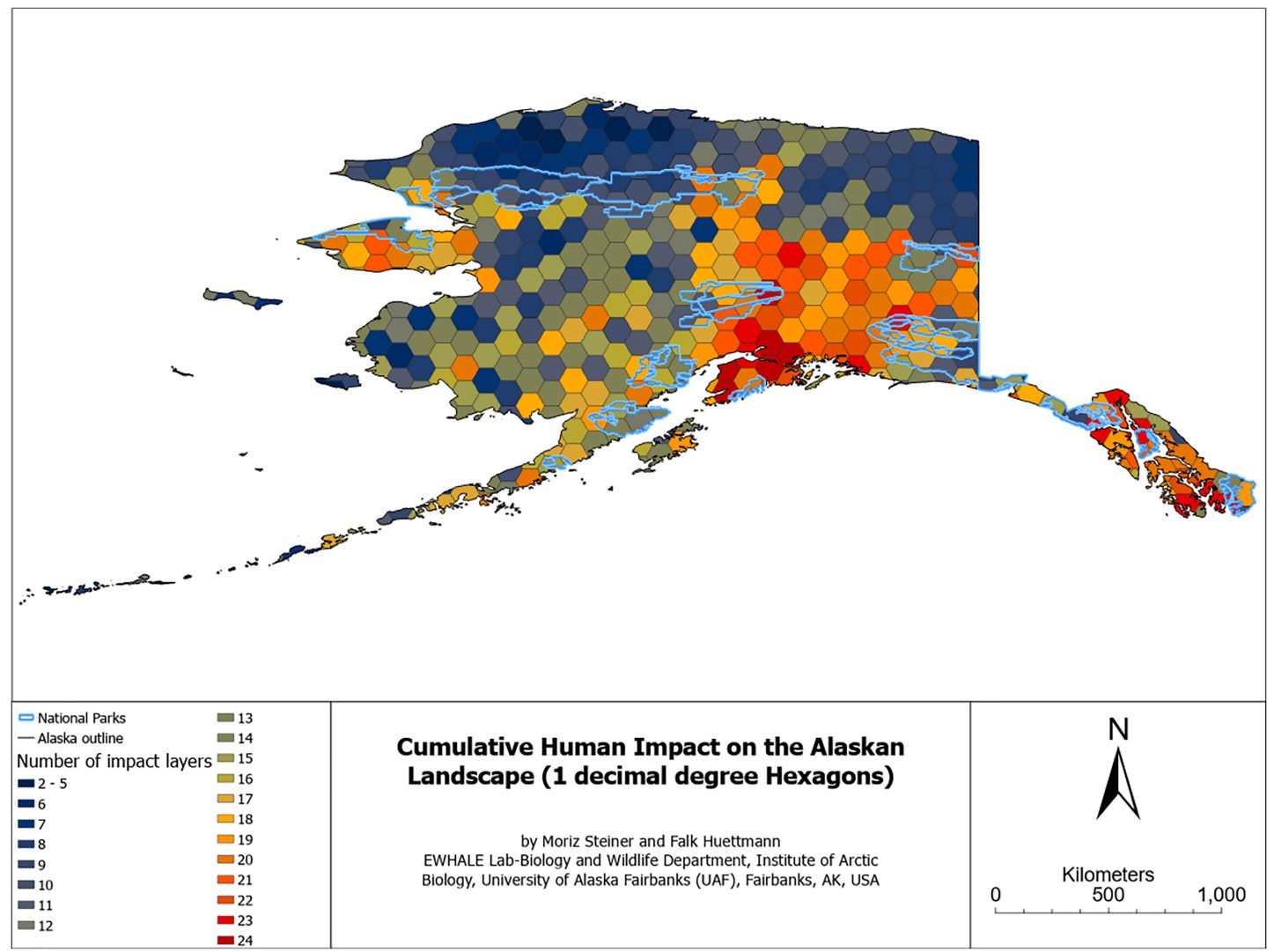

**Figure 6 Cumulative human impact on the Alaskan landscape-hexagons with national park boundaries.**

Homer, Anchorage, Fairbanks, the road system, and Utqiagvik by latitude, longitude, and altitude peaks. With climate change and rising sea levels, these urban hotpots will likely extend and shift (unlikely on a voluntary basis, especially urban hotspots on the coasts; see *Marino, 2015*).

Urban hotspots can be observed in Fig. 8A at longitude −150 decimal degrees for Anchorage, −148 for Fairbanks, −134 for Juneau, and −177 for Adak), in Fig. 8B at latitude 61 for Anchorage, 64 for Fairbanks, 59 for Juneau, 55 for Ketchikan, and 71 for Utqiagvik, and in Fig. 8C at altitude in meters, close to 0 for many coastal villages and Anchorage, approx. 130 for Fairbanks, approx. 400 for Healy, and 6,190 for Denali.

## Qualitative evaluation of the results

Our results show the units and magnitude of human footprint explicit in the coordinate space and elevation, as per the input GIS layers and methods described. As one may expect

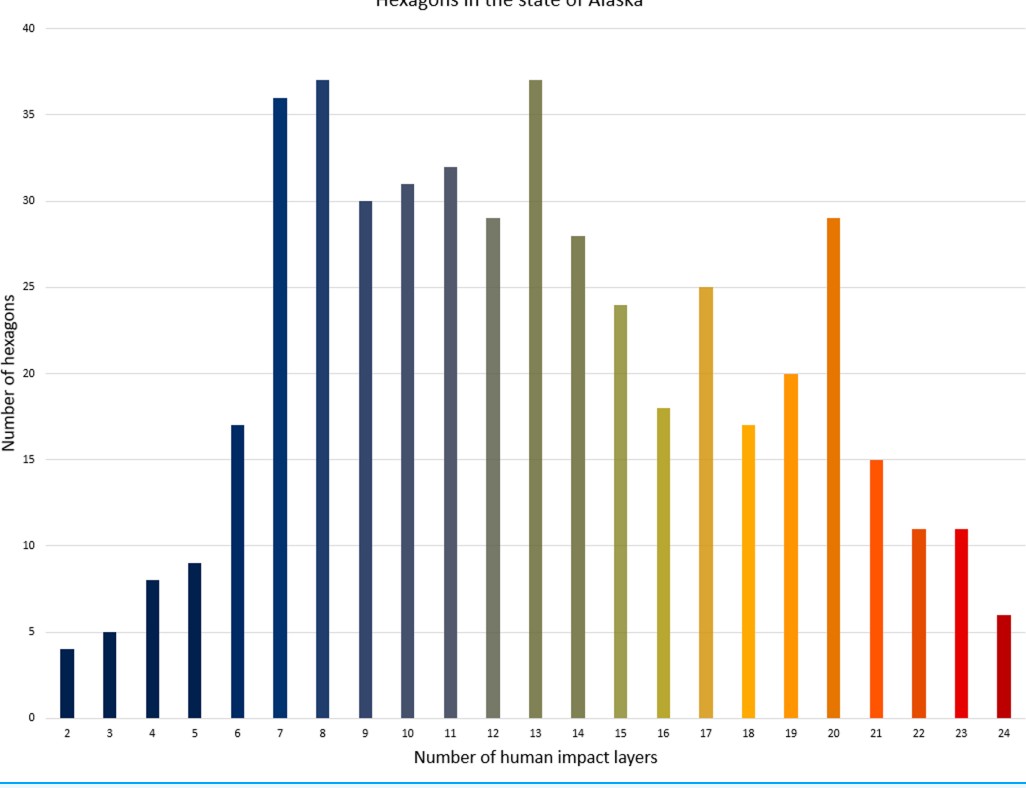

Frequency Distribution of the Number of Human Impact Layers within 1 decimal degree Hexagons in the state of Alaska

**Figure 7 Cumulative human impact on the Alaskan landscape-frequency distribution of hexagons.**

from the Anthropocene (*Dalby, 2016*), from a qualitative perspective, the sheer entirety of the state of Alaska appears to be impacted by human activity in one way or another. Especially when considering atmospheric impacts like man-made $CO_2$ levels rises and methane (GHGs), no area is truly untouched and pristine (see also *Bao, Jia & Xu, 2023* for ecological processes), IPCC reports for national coverage (*Byerly, 2010*), and the lack of summer sea ice on the north shore by 2050 (*Wang & Overland, 2015*). Still, the major urban impact hotspots can currently be found in central and southern Alaska (see Figs. 5 and 6) around major urban areas like the cities of Anchorage, Fairbanks, Juneau, Homer, and Prudhoe Bay, and along major waterways like the Yukon and the Tanana River, as well as major roadways like Highway 1, 2, 3, 4, 6, and 11. Figure 6 additionally displays urban impact hotspots on the Seward Peninsula (central-west).

The National Parks (outlined in light blue) appear to have overall fewer impact hotspots (compared to other areas). However, some also show high human impact hotspots, *e.g.*, Denali National Park (Central Alaska), Bering Land Bridge National Preserve (on the Seward Peninsula), and National Parks in Southeast Alaska. Those include Sitka National Historic Park, Glacier Bay National Park, and Klondike Gold Rush National Historic Park (see details: *National Park Service, 2025*). The most northern National Parks appear to have the lowest human impact hotspots. However, it is noteworthy that some of the major human impact coldspots appear to be outside of the National Parks (*e.g.*, see north of the

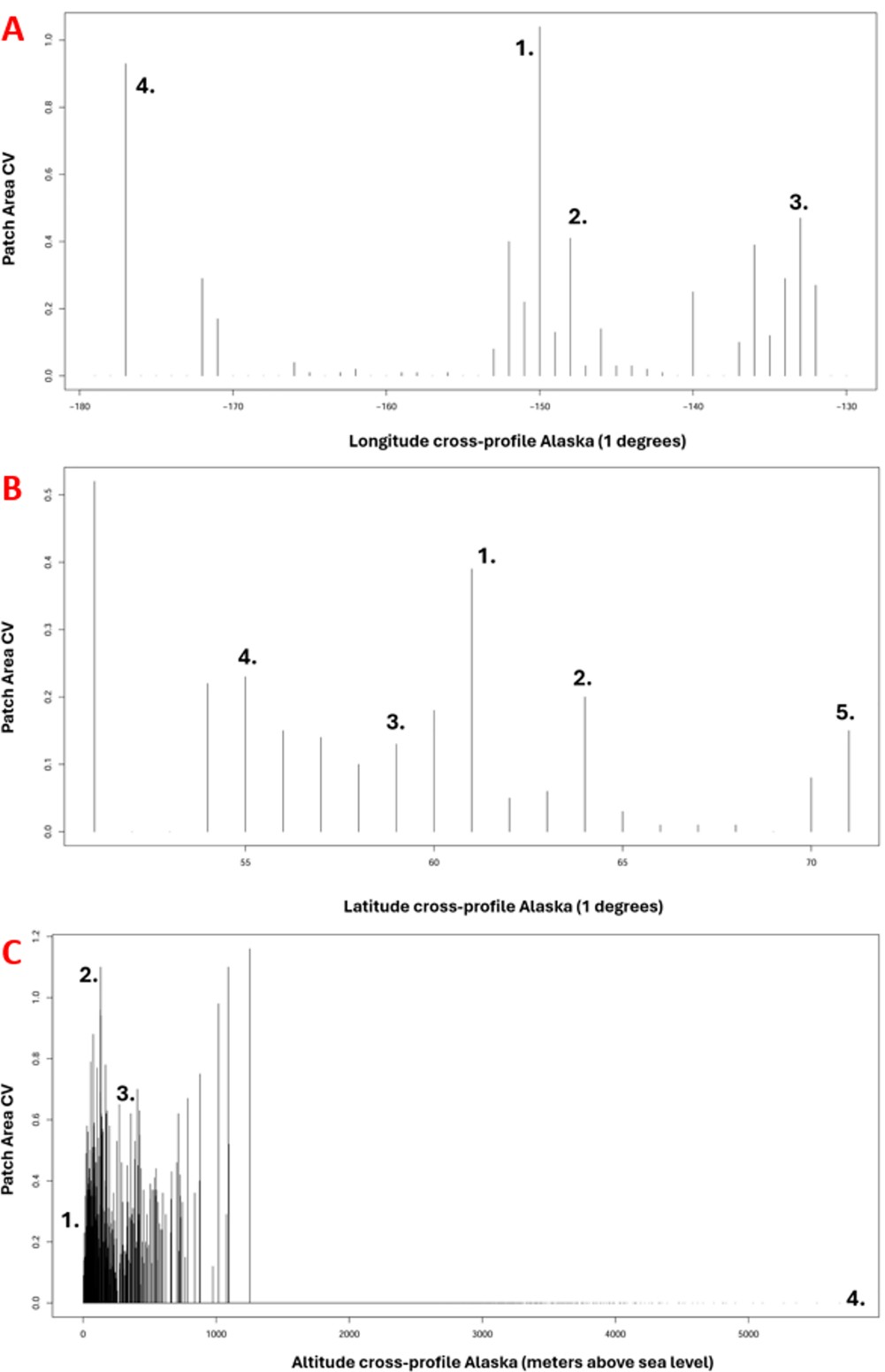

**Figure 8** (A) Longitudinal assessment of urban fragmentation hotspots in Alaska (1 = Anchorage, 2 = Fairbanks, 3 = Juneau, 4 = Adak), (B) Latitudinal assessment of urban fragmentation hotspots in Alaska (1 = Anchorage, 2 = Fairbanks, 3 = Juneau, 4 = Ketchikan, 5 = Ut).

most northern National Parks in Fig. 6, near the Arctic National Wildlife Refuge (on the eastern side) and the North slope on the western side of the Haul Road (Dalton Highway)). Notably, even the blue pixels in Fig. 6 (which can be considered coldspots) can include up to ten human impact layers.

Accordingly, the computed landscape metrics appear as expected, with high edge densities and low patch sizes in highly fragmented urban areas and a low number of total edges in areas with low expected landscape fragmentation. The other included metrics followed this trend.

### Quantitative impact on the Alaskan landscape

The total 30 × 30 m pixel landcover classification map provided by NLCD includes 2,147,483,647 pixels, all urban classes combined add up to a total of 1,433,589 pixels, which only represents 0.067%. To provide a more realistic minimum-estimated human impact on the Alaskan landscape, we summarized all available direct and indirect human impact components presented in the section "Direct and indirect human impact components affecting the Alaskan landscape" using >475 hexagons. In reality, none of the hexagons show 0 (zero) human impact layers and are at least to a small extent impacted by humans as some impact layers cover the whole state (*e.g.*, $CO_2$, Methane, and permafrost melting). Other major land management and governmental activities and acts affect predator control and trapping, as well as the homesteading act in Alaska as the "last frontier" (*Anderson, 2011*; *Boertje, Keech & Paragi, 2010*; *Hull & Leask, 2000*; *Van Ballenberghe, 2006*). Therefore, the true human impact on the Alaskan landscape is most likely close to 100%. A vital consideration supporting this argument is the non-infrastructural impact surrounding human structures. For example, hunting activities are likely to occur within a 1–2 km buffer off all roads and trails, and the same is true for bodies of water as landing pathways for floatplanes. So, even though some remote areas appear to have low human impact, the true human impact on nature and the landscape is prevalent. This finding sets a baseline and has many implications for Alaska, its management, and the U.S. National Park system overall, as well as how Alaska is understood and subsequently managed.

## DISCUSSION

Quantifying human impact on the landscape explicitly in space and time can reveal insightful details about landscape conditions, land use, and intactness. Considering the vast absence of such research *e.g.*, in Alaska, this kind of quantified work is urgently needed for meaningful prioritization and sustainable management (for examples of omission, see, for instance, *Mulder et al., 2023*). The true human impact on the landscape and the overall environment is often underestimated and understated, certainly for Alaska. For instance, the National Land Cover Database (NLCD), which we utilized for the landscape metrics classification, severely underrepresents the urban impact on the landscape (see Figs. 3 and 4). The NLCD map illustrates the most dominant land cover. This means that even if the urban areas only represent a small fraction of a percentage (0.067%) in the data, this only represents the area where this land cover appears to be dominant. As we outlined in this study, the actual area impacted by humans is close to 100%, considering tele-coupling

approaches and atmospheric impacts. Tele-coupling emphasizes feedback signals between distant systems, incorporating coupled systems, such as human and natural systems, social-ecological systems, and human-environmental systems. It connects various distant socioeconomic and environmental interactions and their impacts (*Liu et al., 2019*). Global examples include the Asian Brown Cloud (*Srinivasan & Gadgil, 2002*), air contamination from Norilsk smelter (*Baklanov et al., 2013*), cruise ship industry plumes (*Mendoza-Lara et al., 2023*; *Schulkin, 2002*), NOX brought by oil and gas flaring (*Anejionu et al., 2015*), *etc.* Even when purely focusing on physical impacts the impact is still signification, for example, at least ten impact layers cover 60% of hexagons, and at least ten impact layers cover 90% of hexagons. This pattern–and many of the core fragmentation hotspots–come with an evolution, starting with humans on the landscape at least c. 20,000 years ago (*Fagan & Durrani, 2019*), and then Western contact and colonialism in the 15[th] century (*Gallaher, 2009*), WW1, WW2, oil and gas development in the late 1960s (*Theriot, 2014*), mining (*McNeill & Vrtis, 2017*), and associated boom and bust cycles (*Hollander, 2011*; *Pollock, 2010*). A good example is Anchorage as a colonial port location by J. Cook (*Jones, 2020*) or Fairbanks as a hub for the interior for over 140 years (*Murie, 2003*; *Orttung et al., 2019*). In the year 2025, we find ourselves at the end of globalization; in the Anthropocene and with massive heatwaves promoting the Pyrocene (*Pyne, 2020*, *2022*), with the economy not designed to support species conservation (*Huettmann, 2014*), and with most 'wilderness' species having a direct urban link (*Huettmann et al., 2024*). In the meantime, the military is another major land-holder in Alaska and the U.S., affecting the land and species management as well as access to it on the ground. With a budget higher than most countries' GDP, a sustainable land stewardship would be expected for all lands occupied, managed, and surveillanced by the U.S. military.

Considering that the human impact has not been modeled or presented in a similar format as in this study, it can be considered a starting point from which improvements can be initiated with data made available to do so (including corresponding ISO-compliant metadata; see Appendix 3-https://scholarworks.alaska.edu/handle/11122/15662).

## Data and method limitations

The NLDC map is already pixelated and thus widely inaccurate for its categories, while fragmentation layers dominate in the lack thereof. The limitations of the NLDC map also lie in the missing updates for the state of Alaska, where the contiguous US has received more recent NLDC updates, Alaska has not been updated since 2016. Additionally, with technical advancements since the last update, a usable accuracy, and a higher resolution could also provide additional insights, similar to the latest datasets provided by ESRI (https://livingatlas.arcgis.com/landcover/). Such higher-resolution products still come with the drawback of being composed of multiple tiles rather than a continuous map, which would be a valuable addition to the scientific community and the National Land Cover Database (NLCD).

The traditional Fragstats software is limited by computational aspects such as a modern operating system and a major input file size constraint, hence the reason we attempted to overcome it in this study. This allows for a more realistic assessment as a future reference

to work from. To overcome the computational limitation of the leading landscape ecology analysis software and its metrics, here we successfully show several different approaches to explore pathways that fulfill the computational needs required by this dataset. By exploring several software options, *e.g.*, in-built tools in ArcGIS Pro, ArcGIS, and QGIS (including GRASS GIS), we were able to prove that it is not possible to recreate the results for our large study area that are similar to the ones Fragstats would produce when solely working with the GIS tools outlined above.

RStudio and the 'landscape metrics' package are the only meaningful options that allow computing landscape analyses using larger input files. However, this approach is quite time-consuming. One option to overcome this time-intensive approach is to run the code on a cloud computer. However, that comes with its own constraints and hurdles, including administration, running costs, code modifications, *etc.*

Each landscape metric for the merged urban landcover class took approximately 3–5 days to run on a personal computer (using the R code provided in Appendix 1-https://scholarworks.alaska.edu/handle/11122/15662; and a PC with a 13th Gen Intel i7-13700 processor). This limits the computation of a high number of metrics and landcover classes on a personal computer within a short period of time. For a more exhaustive study and the inclusion of, *e.g.*, other landcover classes and up to 50 landscape metrics, elaborate cloud computing pathways are recommended.

For the cumulative human impact layers, we attempted to compile an inclusive set of data, yet not all the data we initially anticipated to include were digitally available. Table 3 provides an overview of the layers we anticipated to include but were not able to obtain and thereby include in this study. For an even more complete and inclusive study of the actual human impact on the Alaskan landscape, those layers could provide additional insights. Adding archeology layers would greatly help in that effort but none are readily available and on a meaningful scale.

## Closing statement

Alaska lacks a proper quantified digital assessment of the human footprint and impact. Here we present the first workflow and product to do so. From our experience, using R/RStudio is the most effective, yet time-consuming and only realistic approach to analyzing landscape metrics on larger scales with low pixel size and high resolution. The leading software for landscape metrics analysis ("Fragstats") is most efficient for smaller study areas, whereas integrated tools in GIS still contain algorithm problems and bugs and are therefore not recommended at this stage. This project has shown that landscape metrics can be computed for an area as large as the whole state of Alaska with a high-resolution input image of $30 \times 30$ m, as a proof-of-concept. While the overall human footprint was found to be excessive for all of Alaska, it shows a promising outlook for more work on such high-resolution scales and large landscape areas. Future studies can be more exhaustive with the applied methods and analyze all available landcover classes and more landscape metrics. For such scenarios, cloud computing workflows are suggested, where the same R code, as presented here in Appendix 1 (https://scholarworks.alaska.edu/handle/11122/15662), can be utilized.

When looking at the computational outcome, it is clear that the physical human footprint needs to be covered better by the existing best-available official data layers for Alaska and, thus, in public discourse or policy. Looking at a more reality-based and holistic approach, Alaska is far from a pristine state, with a massive impact in just the last three decades–the acceleration–. This reality is not present yet in any wildlife, habitat, or landscape and freshwater water table management across agencies and actors, including the U.S. National Park Service. Relevant policies feature instead the 'originalism' and are widely based on the 'current' and outdated landscape management scheme. In times of man-made climate change, Alaska is not getting more pristine and wilder, nor is the U.S. or its protected areas. The current trend does not indicate a fair and earnest sophistication or apparent vision to reach a sustainable landscape and resource management. Our findings have major implications for native lands, as well as nationally protected areas like the U.S. National Park system and its well-being and meaning. It is of national and international value. With a greater understanding of human impact on the landscape, we as a society can utilize this understanding to manage the Alaskan land more sustainably and potentially set aside land excluded explicitly from direct human impacts, for increased wilderness on this planet.

## LIST OF APPENDICES

Appendix 1: R code for the landscape metrics assessment.

Appendix 2: GeoTIFF files of the computed landscape metrics.

Appendix 3: ISO-compliant metadata of the data and computed results.

Appendix 4: High-resolution versions of sub-figures for Figs. 1, 3, 4, and 8.

All appendices can be accessed *via* the institutional data repository (https://scholarworks.alaska.edu/handle/11122/15662).

### Funding

The authors received no funding for this work.

### Competing Interests

Falk Huettmann is an Academic Editor for PeerJ.

### Author Contributions

- Moriz Steiner conceived and designed the experiments, performed the experiments, analyzed the data, prepared figures and/or tables, authored or reviewed drafts of the article, and approved the final draft.
- Falk Huettmann conceived and designed the experiments, performed the experiments, analyzed the data, prepared figures and/or tables, authored or reviewed drafts of the article, and approved the final draft.

### Data Availability

The code and raw data are available at the University of Alaska data depository: http://hdl.handle.net/11122/15662.

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
