# Peer review of "Moving beyond the physical impervious surface impact and urban habitat fragmentation of Alaska: quantitative human footprint inference from the first large scale 30 m high-resolution Landscape metrics big data quantification in R and the cloud"

_PeerJ, doi:10.7717/peerj.18894_

## Round 0.1 · original submission · Major Revisions

The manuscript promises a comprehensive analysis with ten landscape metrics, but only two metrics (Coefficient of Variation – Patch Area and Edge Density) were computed successfully. This significant reduction limits the study's scope, undermining its ability to capture the full complexity of landscape fragmentation and human impact, and weakens the basis for robust conclusions​

The authors reprojected data and used specific GIS processing steps without explaining the rationale behind these choices or the implications for data accuracy. For instance, they changed coordinate reference systems (CRS) without detailing why this transformation was necessary, which affects data integrity and precision​

The figures are of substandard quality, with low resolution and inadequate labeling, which detracts from the clarity of data visualization and the overall presentation of the manuscript.

The manuscript attempts to quantify “true” human impact by adding various cumulative impact layers (e.g., governance boundaries, game management units), but it fails to explain the relevance or validity of these additions. Many layers seem indirectly related and may overstate human impact without evidence that they affect landscape fragmentation directly

Please address ALL reviewer comments and submit the revised manuscript, if you wish to publish your paper in PeerJ.

Reviewer 1 ·

Basic reporting

1. Removed the keywords that you already used in the title and list them alphabetically.
2. The literature review in introduction section is limited to studies from specific region. Present the newest studies on land use and land cover changes with the focuses on landscape change in other regions.
3. The novelty and overall aims and research questions of the current research should be illustrated at the last paragraph of introduction.

Experimental design

4. Move lines 111 to 119 to study area part.
5. Provide the maps of study area in the world map to highlight the location of the study area.
6. In the study area part: present the extent of the study area (in degree and minutes and second), climate condition, and physiography, percentage area of each land use class, and other information like socio and economic condition.
7. In figure 1, merge the legend with each other. Write the name of each class beside the colorize rectangles.
8. In the figure 1, labeled the name of studied cities on the map.
9. You used National Land Cover Database (NLCD) in the current study which is available until 2016. As land use and land cover changes highly impact by human activities and rapid development, this provided information is too old. You need to use Environmental System Research Institute land use map (ESRI land use map) instead, which has been update until now. This ready map is also high resolution which has been accurately generated.
10. The basic map of the current study is land cover map which has been provided in 2016. Why are you used the sum of CO2 and methane concertation between June 14th 2024 and June 16th 2024. The study period should be consistent.

Validity of the findings

11. Lines 241 to 246 are methodology. You also need to define each landscape metrics.
12. You compared the result of other study in Result part. You need to move this part in to discussion part.
13. Remove the citation in closing statement part.
14. Write the main result for each figures (Quantitative and qualitative).

·

Basic reporting

• Clear and unambiguous, professional English used throughout:
The manuscript is generally well-written in English. However, sections such as the discussion and methods would benefit from additional editing to improve clarity and precision. For instance, certain complex sentences—particularly those that describe methodological steps and interpret landscape metric findings—are sometimes lengthy or contain multiple technical terms in succession. Simplifying sentence structures and breaking down complex information would reduce ambiguity and ensure the intended meaning is fully accessible to readers.
Example from line 122: "We carried out a state-wide landscape analysis of ten landscape metrics for all urban area landcover classes with the 30m resolution National Land Cover Database landcover map as the basis."
Suggested Revision: "We conducted a statewide landscape analysis using ten landscape metrics, based on the 30m resolution National Land Cover Database for urban landcover classes."
Additionally, the sixth paragraph of the introduction effectively contextualizes Alaska's unique environmental and cultural significance. However, removing tangential references, such as U.S. military spending, would allow the focus to remain on Alaska’s natural resources, Indigenous history, and conservation priorities, thereby enhancing coherence.

• Literature references, sufficient field background/context provided:
The introduction effectively establishes a background on human impact on landscapes, particularly within the Alaskan context, and highlights the need for quantified assessments. However, additional citations on recent advancements in landscape metrics and studies on the impacts of climate change on northern ecosystems would further strengthen the context.

• Professional article structure, figures, tables. Raw data shared:
The manuscript generally follows a professional structure, but there is significant room for improvement in map clarity and figure details. Enhancing the design of all maps and providing more comprehensive captions for figures and tables would add valuable context and improve interpretability. For example, adding a brief explanation in the caption for Table 1 about the purpose of including specific layers would help clarify their relevance. Additionally, explaining in the text how each layer contributes to the cumulative human impact analysis would improve scientific clarity.

o Figure 1a: The colors used for land cover classes 12 (Perennial Ice/Snow) and 95 (Emergent Herbaceous Wetlands) appear visually similar, which may lead to confusion. Adjusting the color scheme for clearer differentiation would enhance readability and prevent misinterpretation.

o Figure 1b: Due to the sparse distribution of urban areas, distinguishing between classes at this scale is challenging. Adding a zoomed-in inset of a denser urban area would allow for better visibility and differentiation of urban land cover classes.

o Figures 4b and 4c: The range and threshold values used in Figures 4b and 4c should be clarified to help readers understand the significance of the human impact layers. Providing specific explanations for thresholds (e.g., “more than 10” or “more than 6” impact layers per hexagon) in the figure captions would enhance interpretability.

o Consistency in Figure Labels: Figure 11 should remain labeled as "Figure 4a," while Figure 13 should be re-labeled as "Figure 4c" to avoid confusion and ensure clear reference throughout the manuscript.

• Self-contained with relevant results to hypotheses:
The manuscript is well-structured as a self-contained study, presenting a cohesive investigation into human impact on Alaska’s landscape. The results are aligned with the stated hypothesis, providing relevant data and analysis to support the study’s objectives.

Experimental design

• Original primary research within Aims and Scope of the journal.
The manuscript presents original research that aligns well with the journal’s focus on environmental science and landscape analysis. The research question is clearly defined, relevant, and meaningful, addressing a significant knowledge gap by focusing on high-resolution landscape metrics for assessing human impact in Alaska.

• Research question well defined, relevant & meaningful. It is stated how research fills an identified knowledge gap.
The study’s research question is explicitly stated and well-justified within the introduction. The manuscript effectively identifies the knowledge gap by addressing the limitations of existing datasets, such as the underrepresentation of human impact in traditional sources like the NLCD. The study contributes meaningfully to Alaska's conservation and land management practices, particularly in quantifying and visualizing human impact.

• Rigorous investigation performed to a high technical & ethical standard.
The investigation is conducted rigorously, adhering to high technical and ethical standards in data usage and environmental analysis. The use of publicly accessible Open Access data aligns with industry best practices, promoting transparency and supporting ethical standards in environmental research.

• Methods described with sufficient detail & information to replicate.
While the methods section is detailed, certain areas, particularly those related to data processing for cloud computing and the calculation of individual landscape metrics, would benefit from additional information to ensure full replicability. Providing more explanation about each landscape metric would enhance the scientific clarity and context of the study. Each metric likely serves a specific purpose in analyzing urban fragmentation, and explaining why each was chosen—such as what aspect of landscape structure or pattern it captures and its relationship to human impact—would help readers understand the relevance of each metric to the study’s objectives.
For example, briefly describing metrics such as:

o Edge Density: how it relates to the fragmentation of urban patches,
o Largest Patch Index: its significance in assessing dominant urban areas, and
o Mean Shape Index: its role in understanding the complexity or irregularity of urban patches,

would provide valuable insights for readers, especially those less familiar with landscape metrics.

Validity of the findings

• Robustness and Transparency of Data:
While the study utilizes a large dataset, additional transparency in methodology would significantly enhance the analysis. Expanding on the hexagon analysis—specifically, how the >475 hexagons were employed to calculate human impact—would help readers understand the significance and robustness of this approach. Additionally, clarifying which data layers are categorized as “direct” or “indirect” human impact (e.g., CO₂ emissions, infrastructure, land management activities) would reinforce the comprehensiveness of the results.

• Clarification of Nearly 100% Human Impact Estimate:
The conclusion that nearly 100% of Alaska is influenced by human activity would benefit from further explanation, particularly since some regions may have minimal physical disturbance. Providing additional context here would add credibility to this broad estimate.

• Emphasis on NLCD Limitations:
The discussion acknowledges that the NLCD may underrepresent urban areas, but expanding on specific limitations—such as resolution and classification methods—would help readers understand why urban areas might appear understated, adding rigor to the critique of NLCD data.

• Tele-coupling and Atmospheric Impacts:
A brief explanation of "tele-coupling" and atmospheric impacts would help clarify these concepts for readers less familiar with them. Adding Alaska-specific contexts, such as NOX emissions or regional atmospheric changes, would make this section more directly relevant. Examples like the Asian Brown Cloud add valuable context, but linking these global impacts directly to Alaska’s landscape would create a clearer narrative.

• Clarification on Physical Impact Layers in Figure 4c:
Figure 4c uses thresholds (e.g., “more than 10” or “more than 6” impact layers per hexagon) to represent human influence. These thresholds lack sufficient explanation, which limits interpretability and may hinder readers' ability to fully understand the extent of human influence represented in Figure 4c.

• Implications of Findings for Policy and Conservation:
The discussion mentions implications for Alaska’s management and the U.S. National Park system, but including specific examples of how these findings could influence conservation policies, land-use planning, or resource management would strengthen this section.

Reviewer 3 ·

Basic reporting

The overall presentation of the manuscript is of below-quality quality. There is no proper literature review on the study area. The resolution and presentation of the figures are not up to the mark.

Experimental design

There is no scientific basis for performing the same analysis on a desktop and HPC. It is only a resource support. Only one of these that is enabled to run and complete the analysis can be mentioned. There is also a redundancy in the approach of the study which can also be reduced (more details in the pdf).

Validity of the findings

Only 2 landscape metrics are presented in the current manuscript out of the 10 outlined in the study design. These limited and incomplete results raise concerns regarding the validity and comprehensiveness of the findings. The interpretation of these 02 metrics may not capture the full complexity of landscape patterns as a result of human interventions or provide a robust basis for the conclusions drawn.

Additional comments

I recommend the overall quality improvement of the manuscript presentation, literature review, rationale behind every method, and completion of the suggested study design before communicating it further. If completing all metrics is not feasible, consider revising the study's scope, methods and objectives to present what is possible under current computational support. Appendix 2 is also missing.

Annotated reviews are not available for download in order to protect the identity of reviewers who chose to remain anonymous.

---

## Round 0.2 · accepted · Accept

Thank you very much for your efforts in addressing the comments provided by the reviewers and me. I have carefully reviewed the revised manuscript and your detailed response letter. I am satisfied with the revised manuscript and the efforts made to address the feedback. I am pleased to inform you that your paper is accepted for publication without the need for further external review.

Congratulations on your excellent work, and we look forward to seeing your article published.

Best regards,